# Mouse Models for Application in Colorectal Cancer: Understanding the Pathogenesis and Relevance to the Human Condition

**DOI:** 10.3390/biomedicines10071710

**Published:** 2022-07-15

**Authors:** Chuangen Li, Harry Cheuk-Hay Lau, Xiang Zhang, Jun Yu

**Affiliations:** Institute of Digestive Disease, Department of Medicine and Therapeutics, State Key Laboratory of Digestive Disease, Li Ka Shing Institute of Health Sciences, CUHK Shenzhen Research Institute, The Chinese University of Hong Kong, Hong Kong, China; lcg20050502@aliyun.com (C.L.); harrylau@link.cuhk.edu.hk (H.C.-H.L.); jenniferzhang@link.cuhk.edu.hk (X.Z.)

**Keywords:** colorectal cancer, mouse model, carcinogen, transgenic, treatment

## Abstract

Colorectal cancer (CRC) is a malignant disease that is the second most common cancer worldwide. CRC arises from the complex interactions among a variety of genetic and environmental factors. To understand the mechanism of colon tumorigenesis, preclinical studies have developed various mouse models including carcinogen-induced and transgenic mice to recapitulate CRC in humans. Using these mouse models, scientific breakthroughs have been made on the understanding of the pathogenesis of this complex disease. Moreover, the availability of transgenic knock-in or knock-out mice further increases the potential of CRC mouse models. In this review, the overall features of carcinogen-induced (focusing on azoxymethane and azoxymethane/dextran sulfate sodium) and transgenic (focusing on *Apc*^Min/+^) mouse models, as well as their mechanisms to induce colon tumorigenesis, are explored. We also discuss limitations of these mouse models and their applications in the evaluation and study of drugs and treatment regimens against CRC. Through these mouse models, a better understanding of colon tumorigenesis can be achieved, thereby facilitating the discovery of novel therapeutic strategies against CRC.

## 1. Introduction

Colorectal cancer (CRC) is the third most common cause of cancer-related death worldwide [1]. CRC results from the progressive accumulation of genetic and epigenetic alterations that lead to the transformation of normal colon mucosa to adenocarcinoma. Approximate 90% of CRC cases are sporadic and occur in patients without genetic predisposition or family history of CRC [2], whereas around 2–5% of CRC cases are hereditary [3]. The two most common inherited CRC are hereditary nonpolyposis CRC and familial adenomatous polyposis (FAP). To date, the molecular mechanism in different stages of CRC development remains unclear. Thus, it is important to investigate the molecular initiation and progression of colon tumorigenesis using preclinical animal models.

The molecular mechanism of CRC progression has been identified since the establishment of the *Apc*^Min/+^ transgenic mouse model. Genetic mutation of adenomatous polyposis coli (*APC*) has been detected in 70% of patients with sporadic CRC [4], indicating a crucial role of *APC* mutation in colon tumorigenesis. Consistently, preclinical studies reported that *Apc* mutation can induce spontaneous formation of colon tumors in mice [5]. The *Apc*^Min/+^ transgenic mouse model is thus commonly used for research on colon tumorigenesis [6]. Meanwhile, deleting several other genes such as *p53*, Kirstein rat sarcoma viral oncogene homolog (*KRAS*), and phosphatase and tensin homolog (*PTEN*) have also been reported to contribute CRC development [7], although not all of these genetic alterations are similar to *Apc* mutation, which can induce spontaneous colon tumorigenesis. However, carcinogens are also widely applied to mice for studying CRC development. Azoxymethane (AOM) is the most used carcinogen to mimic the development of sporadic CRC [8,9]. Co-administration of AOM and dextran sulfate sodium (DSS) is also a common approach to induce colon tumorigenesis in mice, recapitulating the pathogenesis of colitis-associated CRC (CAC) [10].

The mechanism of CRC development is complex, since both genetic and environmental factors are involved [11]. Although in vitro cell cultures are efficient, they cannot accurately recapitulate the physiological conditions in human CRC. Hence, mouse models are necessary tools for preclinical studies of CRC. In this review, we summarize the overall features of the most commonly used carcinogen-induced (AOM and AOM/DSS) and transgenic (*Apc*^Min/+^) mouse models of CRC, with further elucidation on their mechanisms to induce colon tumorigenesis. We further evaluate limitations of these mouse models and how their applications in preclinical studies can facilitate clinical benefits.

## 2. Carcinogen-Induced Mouse Model of Colorectal Cancer

To date, a variety of carcinogen-induced CRC mouse models has been established: (1) AOM, methylazoxymethanol and 1,2-dimethylhydrazine (DMH); (2) heterocyclic amines including 2-amino-1-methyl-6-phenylimidazo and 2-amino-33-methylimidazo [4,5-f] quinoline; (3) aromatic amines including 3,2-dimethyl-4-aminobiphenyl; and (4) alkylating compounds including methylnitrosourea and N-methyl-N-nitro-N-nitrosoguanidine [6]. Among them, AOM is the most commonly used carcinogen to induce CRC in mice. Meanwhile, the combined application of AOM and colitogen DSS is also widely considered as another robust method to induce CRC in mice. In this section, the overall features of AOM and AOM/DSS mouse models as well as the mechanisms of how these carcinogens induce colon tumorigenesis are discussed.

### 2.1. AOM- or AOM/DSS-Induced Mouse Model of Colorectal Cancer

AOM is the metabolite of DMH. Compared to other carcinogens, AOM is more efficient in inducing colon carcinogenesis attributed by its high stability [12,13]. AOM specifically induces tumor formation in the colons of mice. Given by its high efficiency and reliability, AOM remains one of the most commonly used carcinogens to induce colon tumorigenesis in mice. To date, most studies provide a total of six AOM doses (10 mg/kg intraperitoneal injection per week), and mice can develop colon tumors about 24 weeks after receiving the last AOM injection [14] (Figure 1A). 

The relationship between inflammation and CRC has been well established since Crohn’s colitis and ulcerative colitis were recognized as being associated with increased risk of CRC [15]. In 2003, the AOM/DSS mouse model was the first reported to mimic CAC with 100% tumor incidence in the distal colon [10]. Currently, the most widely used approach is to intraperitoneally inject AOM (10 mg/kg body weight) once, followed by three treatment cycles of 1–3% DSS in drinking water for 5 days, and then regular water treatment for 14 days [16,17] (Figure 1B). Notably, the efficiency of tumor induction and tumor number depend on the dosage of AOM and/or concentration of DSS, which have greatly varied among CRC studies. 

### 2.2. Sensitivity to Carcinogen and Tumor Characteristics in Different Mouse Strains

Mice with different genetic backgrounds have distinct sensitivity to AOM. Rosenberg et al. showed that the sensitivity to AOM is varied among different mouse strains [6]. A/J and SWR/J mice have great sensitivity to AOM with a high incidence of colon tumors. C57B/L6 and Balb/c mice have moderate sensitivity with relatively less incidence of colon tumors compared to A/J and SWR/J mice, whereas AOM administration in AKR/J and 129/SV mice cannot induce any formation of colon tumors [6]. The authors also evaluated features of AOM-induced tumors in different mouse strains. The tumor morphology was found to be similar among these mouse strains, while metastasis or invasion was not observed even in the mouse strain with high AOM sensitivity. AOM-induced tumors are exclusively located at the distal region of the mice colon. In A/J and SRJ/R mice, which are the strain with highest AOM sensitivity (develop ≥ 10 tumors per mouse within 8 weeks after AOM treatment), their colons are covered by multiple coalescing tumors with rectal bleeding, while C57BL/6 mice that have moderate AOM sensitivity can develop up to 5.5 tumors per mouse [6,18]. Moreover, histological features are also similar among mouse strains of which the intramucosal and expansile tumors are hypercellular with closely packed colon epithelial cells [19]. Tumor crypts are all composed of closely packed cells with indistinct cell borders, regardless of mouse strains. Infiltration of neoplastic cells into the muscular wall of the distal colon can also occur [19]. Contrastingly, although A/J and SWR/J mice have similar AOM sensitivity, histopathological progression varies among these two mouse strains. Tumors in SWR/J mice can develop features of carcinoma in situ with epithelial crypts and inter-glandular stroma 8 weeks after AOM treatment, while A/J mice have less tumor progression at 8 weeks of AOM treatment but with significantly larger tumor size [18]. No metastasis and invasion were observed even in the mouse strain with the highest AOM sensitivity [6], indicating that this model is suitable to study early-stage but not late-stage or metastatic CRC.

Similar to AOM, mice with different genetic backgrounds also have distinct sensitivity to AOM/DSS. A previous study treated four different mouse strains (BALB/c, C3H/HeN, C57BL/6N and DBA/2N) with AOM (10 mg/kg body weight) and 1% DSS [20]. The incidence of colon adenocarcinoma was reported to be 100% in BALB/c and 50% in C57BL/6N, but none was detected in the other two strains. Inflammation seems to be independent from the development of colon adenocarcinoma after AOM/DSS treatment, of which C3H/HeN has the most severe inflammation, followed by Balb/c and DBA/2N, whereas C57BL/6N has the least amount of inflammation, even though colon adenocarcinoma is present in this mouse strain. Moreover, the score of nitrotyrosine positivity was found to be Balb/c > C57BL/6N > C3H/HeN > DBA/2N. These results therefore indicate that the difference in sensitivity to AOM in distinct mouse strains can be attributed to responses toward nitrosative stress caused by DSS-induced inflammation [20].

In general, tumors developed in AOM/DSS-treated mice can recapitulate the main histopathological features of human CRC such as the predominant location of tumors in the middle and distal colon [10]. These tumors were histologically characterized as tubular adenoma or were well or moderately differentiated tubular adenocarcinoma [10]. AOM/DSS-treated mice show persistent colitis with diarrhea and colon bleeding, while there is no significant difference in colon length compared to untreated control mice [10]. In AOM/DSS-treated mice, colon tumors exhibit signs of colitis characterized by crypt architectural distortion and lamina propria inflammation, while a few spotted ulcers with regenerative changes can also occur on the colon mucosa [21]. Moreover, tumor invasion to submucosa, muscularis propria or serosa was observed in AOM/DSS-treated mice [10]. Importantly, these findings demonstrated that unlike AOM-induced tumors that are mostly adenoma, AOM/DSS treatment can induce formation of the full process of colon tumorigenesis, progressing from initial crypt proliferation to the final development of colon carcinoma.

### 2.3. Modeling Different Stages of Colon Tumorigenesis

Aberrant crypt foci (ACF) are putative preneoplastic lesions in the colon, and they have been observed in patients with CRC and in patients with FAP [22]. Previously, ACF were used as a short-term bioassay to evaluate the role of nutritional components at the early stage of colon tumorigenesis, even though ACF are not directly related to the early formation of tumors [23]. Two types of ACF have been characterized: classical elevated ACF whose small crypts are elevated from the surrounding epithelium and flat ACF without elevated structure. AOM can induce flat ACF with similar morphology and expressions of β-catenin and cyclin D1 as in CRC tumors, indicating a continuous progression from monocryptal dysplastic ACF to tumor formation, whereas these features were not observed in classical elevated ACF [23]. Moreover, flat ACF grows significantly faster than classical elevated ACF [23], further implying the close relationship between colon tumorigenesis and flat ACF instead of classical elevated ACF (Figure 1C). Given its association with the formation of flat ACF, AOM treatment is partially able to model the early stage of CRC development in mice.

As AOM alone can induce adenomas only, a novel transgenic mouse model with AOM treatment that can develop colon carcinoma to recapitulate sporadic CRC was recently established. CDX2P-Cre-*Apc*^+/LoxP^ is first constructed, where the induction of CDX2P-Cre can help to heterozygously delete the *Apc* gene in mice colon epithelium. AOM (7.5 mg/kg body weight) is then injected into 10-week-old CDX2P-Cre-*Apc*^+/LoxP^ mice, and these mice can develop 4–5 colon tumors at week 15 with no visible tumors in the small intestine [24]. Histopathological examination showed that colon tumors in these mice can range from well-differentiated adenomas to invasive adenocarcinomas and can exhibit many dominant characteristics of human colorectal adenocarcinomas such as increased nuclear-to-cytoplasmic ratio, heterochromatin and back-to-back gland formation, and tumor invasion into the muscularis mucosa. These observations therefore indicate that this newly established mouse model is capable of recapitulating human sporadic CRC. 

### 2.4. Molecular Mechanisms of Carcinogen-Induced Colon Tumorigenesis

The activation of oncogenes and loss of tumor suppressor genes play important roles in human CRC development, as well as in carcinogen-induced tumor development. AOM-induced and AOM/DSS-induced colon tumorigenesis have similar mechanisms such as the activation of oncogenes (e.g., β-catenin) and inactivation of tumor suppressor genes (e.g., *Apc* and *p53*), probably due to the application of AOM in both models. Meanwhile, some reports showed the difference in molecular mechanisms between these two models, especially *Kras* and cyclooxygenase 2 (COX2). In general, each stage of carcinogen-induced colon tumorigenesis involves several functional pathways, for instance, *Apc*/β-catenin, *Kras*, c-Myc and global hypermethylation are activated in the early stage, while COX2 and iNOS are activated in the progression from adenoma to carcinoma [25] (Figure 2A).

*KRAS* is one of the most prominent proto-oncogenes in colon tumorigenesis belonging to the RAS family, and it controls various biological processes including survival, growth, proliferation, differentiation and apoptosis [26]. Activation of *KRAS* is associated with various oncogenic pathways including PI3K/AKT/mTOR signaling to promote proliferation and to suppress apoptosis of tumor cells [27]. In human CRC, 42.4% of patients involve *KRAS* mutation in codons 12 and 13 [28], while the frequency of *Kras* mutations in AOM-treated mise was reported to be 62% [29]. In contrast, the *Kras* mutation is not always present in the colon tumors of AOM/DSS-treated mice [30], consistent with the fact that *Kras* mutation less frequently occurs in colitis-associated colorectal tumors than sporadic tumors [31].

β-catenin, an intracellular signal transducer in the downstream of Wnt signaling, was found to be frequently activated in the colon of AOM-treated mice. β-catenin is involved in colon tumorigenesis under mutated or overexpressed status with a mutation in codons 33, 34, 37, 41, being G:C to A:T transitions in the tumors of AOM-treated mice [32]. β-catenin was also found to be activated in the colon of AOM/DSS-treated mice, while inhibition of β-catenin can protect the colon from AOM/DSS-induced tumorigenesis [25]. AOM can also cause β-catenin intracellular translocation from the cytoplasm to nucleus, leading to its activation [33]. These results collectively suggest that AOM and DSS induce hyperactivation of the oncogenic Wnt/β-catenin signaling, thereby accelerating colon tumorigenesis.

COX2 is crucial in regulating inflammatory response and lipid metabolism. It is well known that inflammation is a mediator of colon tumorigenesis [34], whereas altered lipid composition can disrupt intestinal epithelial barrier function [35]. Given its functional roles, COX2 can be associated with inflammation and lipid metabolism to contribute to colon tumorigenesis. It was reported that the COX2 mutation is undetected in the early stage of CRC, whereas in large colon adenocarcinoma, the incidence of COX2 mutation is around 80%, indicating that the COX2 mutation is pivotal in the late stage of CRC [36]. Ishikawa et al. found that COX2 is markedly increased in colon tumors, compared to the normal area of the colon in AOM/DSS-treated mice [37]. However, they also showed that COX2 is not required in CAC development, as the expression of COX2 is mainly located in tumor-infiltrating macrophages, fibroblasts, and endothelial cells instead of epithelial cells. Meanwhile, CAC can still be developed in COX2 knock-out mice after AOM/DSS treatment, thus demonstrating that COX2 mutation may not be essential for AOM/DSS-induced colon tumorigenesis. In comparison, although tumors in AOM-treated mice also have significantly overexpressed COX2, AOM treatment but without DSS administration fails to induce any colon tumors in COX2 knock-out mice [37]. These results indicate the distinct roles of COX2 in AOM/DSS-induced CAC and AOM-induced sporadic CRC, of which COX2 is only essential for AOM-induced colon tumorigenesis. Moreover, much more oncogenic pathways were reported to be activated after AOM or AOM/DSS administration. For example, AOM can enhance the activity of EGFR Tyr-k and proto-oncogene c-Myc to initiate colon carcinogenesis [9,38]. Other studies revealed that the Sphk1/S1P/S1PR1 axis plays a crucial role in CAC development, while using FTY720, a Sphk1 inhibitor, can significantly reduce colon tumorigenesis in AOM/DSS-treated mice [39,40,41] (Figure 2B).

In addition to oncogenes, AOM also affects tumor suppressor genes to promote colon tumorigenesis. In 80–90% of CRC patients, the initial step of tumorigenesis is the loss of tumor suppressor gene *APC* [9], while in mice, AOM induces *Apc* mutation, including downregulating its expression and shortening the length of *Apc* [42,43]. The loss of *Apc* was also observed in the colon tumor of mice given AOM/DSS. Kishimoto et al. demonstrated that inhibition of COX2 can increase the expression of *Apc* in the colon of rats treated with AOM/DSS, indicating an inverse correlation between *Apc* and inflammation [44]. These results suggest that the loss of *Apc* is associated with inflammation, a risk factor of colon tumorigenesis. The direct interplay between *Apc* and COX2 in AOM/DSS-induced CAC development remains unclear.

Another important tumor suppressor gene affected by AOM is *p53*, which has increased sensitivity to AOM under null status [45]. However, unlike *Apc*, the expression of *p53* is increased after AOM treatment but with reduced activity [46]. Mechanistically, AOM-induced COX2 upregulation can interact with *p53* to inhibit *p53*-dependent transcription [47]. Another study also revealed that COX2 products can cause *p53* accumulation in the cytosol to inhibit *p53*-dependent apoptosis [48]. These results indicate that the suppressed activity of *p53* is associated with COX2 upregulation induced by AOM (Figure 1C). Similarly, *p53* was also found to be elevated in AOM/DSS-treated mice [49]. Since DSS alone cannot upregulate *p53* [17], the increased expression of *p53* may result from AOM treatment. Altogether, carcinogen treatment can induce molecular alterations at the genetic level, contributing to colon tumorigenesis and CAC development.

### 2.5. Gut Microbiota Is Pivotal for Carcinogen-Induced Colon Tumorigenesis

Recent studies showed that the gut microbiota play a pivotal role in AOM-induced CRC mouse models. Colon tumor formation in AOM-treated germ-free IL10^−/−^ mice is greatly abrogated compared to conventional AOM-treated IL10^−/−^ mice due to the impaired inflammatory response [50]. Our previous study also demonstrated that antibiotics administration markedly reduces AOM-induced colon tumor number in mice fed with a high-fat diet [51]. It is well known that inflammation is important for colon tumorigenesis [15]. Bacteria invasion into epithelial cells is essential for inflammation to occur, since antibiotics administration can inactivate ATF6 and STAT3 signaling due to the lack of bacteria invasion [52]. Furthermore, the microbial community within a colorectal neoplasia can be heterogenous, and such heterogeneity is associated with colon tumorigenesis [53]. Moreover, single-cell sequencing revealed the distinct signature of immune cells in different regions of colon tumors [54], which may be associated with the heterogeneous intratumoral microbiota. These findings indicate that bacteria-mediated inflammation is a key step in the initiation of AOM-induced colon tumorigenesis (Figure 1C).

## 3. Transgenic Mouse Model of Colorectal Cancer

In addition to the carcinogen-induced CRC mouse model, transgenic mice are also widely used to study colon tumorigenesis. In particular, as *APC* mutation occurs in over 90% of CRC patients, different transgenic mice have been established with *Apc* mutation being the centrality of these models. In this section, we explore the overall features of the *Apc* mutant mouse model as well as other transgenic mice for the study of spontaneous colon tumorigenesis.

### 3.1. Apc Mutant Mouse Model of Colorectal Cancer

As *Apc* mutation is considered to be the initial step in FAP progression, many mouse models have been developed to mimic FAP by carrying *Apc* mutation (Figure 3A). In 1990, the first *Apc* mutant mouse model was generated by mutagen N-ethyl-N-nitrosourea, leading to the mutant, multiple intestinal neoplasia (Min), which carries a truncation mutation at codon 850 [9,55]. Heterozygotic *Apc*^Min/+^ C57BL/6 mice can develop over 100 adenomas in the small intestine with a few adenomas developed in the colon [56]. *Apc*^Min/+^ mice have many genetic and phenotypic similarities to patients with FAP, with 90% identical in orthology. However, *Apc*^Min/+^ mice mostly develop intestinal adenomas in contrast to patients with FAP, which predominantly develop colon lesions. A histological study showed that colon tumors generated from the *Apc* mutation are benign adenomas [57]; hence, this model is suitable for studying the early stage of CRC instead of the late stage (Figure 3A). Using gene knock-out approaches, several *Apc* mutations in other loci have been generated. Mice carrying *Apc*^Δ716^, *Apc*^Δ14^ or *Apc*^1638N^, containing truncating mutations at codon 716, 14 or 1638, respectively, were shown to have varied tumor numbers, indicating the difference in sensitivity among different mutation loci of the *Apc* gene [9,38]. *Apc*^1638N^ seems to be insensitive to *Apc* mutation, as mice carrying *Apc*^1638N^ only develop a few tumors in the small intestine and no tumors in the colon, while tumor numbers in the small intestine and colon of mice carrying *Apc*^Δ716^ and *Apc*^Δ14^ are significantly higher. Nevertheless, although more mouse models targeting the *Apc* gene have been developed, *Apc*^Min/+^ remains the most commonly used transgenic mouse model of CRC [38].

Gene functional studies have identified numerous genes that impact CRC progression. The application of Cre-Loxp makes it possible to delete or overexpress a gene in a specific organ. By combining Cre-Loxp gene modification with *Apc*^Min/+^ mice, various mouse models with multiple genetic mutations can be established. LGR5, which is a G-protein coupled receptor binding R-spondin to enhance WNT signaling, was shown to be a marker of intestinal stem cells (ISCs) [58]. Using Cre-Loxp to delete *Apc* in mice Lgr5^+^ ISCs leads to rapid formation of intestinal adenomas, strongly indicating that Lgr5^+^ ISCs are the cells of origin for intestinal cancer. Moreover, *Apc*^Min/+^ and *Apc*^Δ716^ only induce mostly the formation of adenomas, but not invasive tumors, whereas mice carrying combined *Apc*^Min/+^ with *Smad3*^−/−^ as well as *Apc*^Δ716/+^ with *Smad4*^+/−^ can develop colon adenocarcinoma [59,60]. Another study explored the impact of *p53* on colon tumor invasion by utilizing Cre-Loxp to generate *Apc*^fl/+^*p53*^fl/+^ and *Apc*^fl/+^*p53*^R172H/+^ mice, which show 25% and 100% stroma invasion, respectively [61]. These results therefore indicate the tumor-suppressive role of *p53* as well as the diverse impacts of *p53* with different mutant locus. For oncogene *Kras* in which its mutation occurs in early CRC, mice with both mutant *Kras*^G12V/+^ and *Apc* develop invasive intestinal carcinoma, while mice with mutant *Kras*^G12V/+^ alone need a much longer duration (>500 days) to develop carcinoma [62]. Collectively, these findings potentiate the importance of the *Apc*^Min/+^ mouse model in the study of colon tumorigenesis by demonstrating the initiative role *Apc* mutation in CRC development.

Recent studies show that the gut microbiota play pivotal roles in CRC initiation and progression. The enrichment of enterotoxigenic *Bacteroides fragilis* and colibactin-expressing *Escherichia coli* was observed in patients with FAP, which were also reported in *Apc*^Min/+^ mice [63,64], further indicating that the *Apc* mutant mouse model can accurately recapitulate human FAP. Meanwhile, *Apc*^Min/+^ mice fail to develop colon tumors under germ-free condition due to their abolished inflammatory response [65]. However, when these mice are transferred to a specific pathogen-free environment, colon tumors can be reintroduced. Hence, these findings indicate that tumor formation in *Apc*^Min/+^ mice needs the presence of an intact gut microbiota.

### 3.2. Molecular Mechanism of Apc Mutation-Induced Colon Tumorigenesis

Tumors in *Apc*^Min+^ mice are mainly adenomas that usually have either polypoid or crateriform appearance with compression or adjacent tissues. The mechanism of how *Apc* mutation induces the formation of colon adenoma has been explored for a long time. In general, the *Apc* gene has four binding sites: β-catenin binding site, end-binding protein 1 (EB1) binding site, microtubule binding site, and discs-large (DLG) binding site [66] (Figure 3B).

β-catenin is involved in the destruction complex together with APC, Axin and GS3K, and this complex is important for WNT/β-catenin signaling [9]. Normally, the destruction complex is inactive with β-catenin maintained at a low level, whereas alteration of this complex can lead to tumorigenesis [67]. Accumulated studies have revealed that β-catenin is activated with knock-out or mutant *Apc* [68], driving the increased level of β-catenin and promoting colon tumorigenesis (Figure 3B). 

A recent study showed that EB1 expression is increased in human tumor samples, whereas *APC* expression in the same sample is reduced [69]. This inverse relationship between *Apc* and EB1 is also present in murine models, of which a previous study reported that EB1 expression is two-fold higher in rats containing the germline *Apc* mutation, compared to wild-type controls [70]. Mechanically, EB1 was identified to be associated with *Apc* carboxyl terminus [69]. The *Apc*-EB1 complex can connect microtubule spindles with kinetochores to regulate microtubule stability [69]. Given its involvement in cell cycle regulation, EB1 is considered to play a crucial role in colon tumorigenesis (Figure 3B).

DLG is located at cell–cell contacts where it acts as a scaffold to interact with other proteins [71]. DLG has been identified as a tumor suppressor essential for regulating cell polarity and proliferation [72]. DLG can form a complex with *Apc*, and this complex plays a crucial role in suppressing cell proliferation by blocking cell cycle progression from the G0/G1 to S phase [71], whereas these anti-tumorigenic activities can become weaker if *Apc* is mutated or truncated [71]. The ability of DLG to disturb the transformation of cell growth is also hampered by β-catenin. Subbaiah et al. found that overexpressed β-catenin drives tumorigenesis by enhancing DLG degradation, while ablated β-catenin can raise the stability of DLG [72]. These results collectively indicate that the altered activity of DLG induced by *Apc* mutation can contribute to colon tumorigenesis (Figure 3B).

Besides genetic modification, recent studies also reported that *Apc*^Min/+^ mice exhibit extensive aberrant DNA methylation in intestinal adenoma, which is a hallmark of CRC [73]. Homozygous mutation of *Apc* can lead to decreased DNA methylation at the promoters of genes implicated in intestinal cell fate specification, such as hoxd13a and pitx2, thereby contributing to CRC development [74]. The *Apc* mutation also affects the DNA demethylase system in mice, including cytidine deaminases Aid and Apobec2a, thymine glycosylase Mbd4, and DNA repair protein Gadd45α, whereas all these epigenetic alterations were observed in human adenoma with germline *Apc* mutation [75]. Meanwhile, the *Apc* mutation can alter the gut microbiota, and furthermore, retinoic acid is metabolite derived from gut bacteria [76,77]. Hence, it is possible that mutant *Apc* alters the gut microbiota and their metabolites to modulate DNA methylation. Taken together, these results indicate that *Apc* loss can induce DNA methylation to facilitate the initiation of CRC development (Figure 3B). 

### 3.3. Cre-Loxp-Based Mouse Models of Colorectal Cancer

Since the establishment of Cre-Loxp technology in the 1990s, deleting any genes in any tissue of interest has become plausible, of which Cre can mediate the recombination of Loxp sites. To date, tamoxifen-inducible Cre-ERT2 is the most commonly used approach to generate conditional knock-out or knock-in mice. Generally, mice carrying tissue-specific inducible Cre-ERT2 is inter-crossed with mice carrying the target gene, which is flanked by Loxp recombination sites (Figure 4). The Cre enzyme is then activated by tamoxifen administration, resulting in conditional addition of the target gene in specific tissue [78]. This technology has been frequently applied to mouse models as aforementioned. Mice carrying the knock-out or knock-in gene of interest can be treated with AOM or AOM/DSS or can be inbred with *Apc*^Min/+^ mice to determine the effect of a specific gene in CRC development and therapeutics.

ATF6 is a basic leucine zipper transcription factor belonging to the CREB and ATF family of transcription factors [79]. Upon endoplasmic reticulum stress, ATF6 induces the transcription of genes encoding chaperones and enzymes that facilitate protein folding and maturation. Previously, ATF6 activation was reported in CRC patients [80], but its exact role in CRC development is still unclear. To address this question, a Villin-Cre Rosa 26 nATF6^fl/fl^ mouse model was established with the overexpression of activated ATF6 specifically in intestinal epithelial cells [52]. These mice spontaneously develop adenomas in the large intestine within 12 weeks. Similar to AOM-treated and *Apc*^Min/+^ mice, tumor formation in this transgenic mouse model also needs the presence of an intact gut microbiota [52]. In addition, CyCAP is a widely expressed secreted glycoprotein that modulates host response to bacterial endotoxins [81]. A previous study identified that CyCAP is a murine orthologue of the TAA90K/Mac-2-binding protein, which can suppress endotoxin signaling in the colon mucosa of mice [82]. In contrast, CyCAP^−/−^ mice spontaneously develop colon mucosal hyperplasia within 8 weeks, indicating that this transgenic mouse model is suitable to study the early stage of colon tumorigenesis.

## 4. Colorectal Cancer Metastasis Mouse Models

Metastasis is a main cause of death for patients with CRC; hence, utilizing mouse models to recapitulate the clinical characteristics is crucial for studying the underlying mechanism and for developing effective treatment against metastasis. Although substantial CRC mouse models have been established, models that can develop features of metastasis remain rare. Here, several mouse models of metastasis are discussed. 

### 4.1. Cell and Organoid Xenotransplantation

The xenograft model refers to the injection of cancer cell lines into mice. Xenograft transplantation readily induces tumor invasion and metastasis, which depends on the site of inoculation. Subcutaneous injection of cancer cells is commonly used in xenograft models due to its convenience and high success rate to induce tumor formation, yet it fails to produce metastasis, while orthotopic injection of CRC cells into specific organs such as caecum, tail vein, and spleen can lead to metastasis in liver, lung, and bones [83]. In general, to evaluate metastasis, human cells with ectopic expression of an oncogene are injected to nude mice or mice with severe combined immunodeficiency (SCID). For example, a previous study injected the CRC cell line HCT116 with ectopic *RNF6* expression into nude mice and reported a 100% incidence rate of lung metastasis in these xenograft mice [84], indicating that *RNF6* can promote the invasion and migration of CRC cells. Notably, a major problem of xenograft models is the incomplete tumor microenvironment, of which the tumor barrier such as the basement membrane is lacking in xenograft mice. Moreover, as nude and SCID mice are immunocompromised, the xenograft model with the injection of cancer cell lines cannot accurately recapitulate metastasis in human patients.

To overcome the problems of xenograft mice with cell transplantation, another patient-derived orthotopic xenograft (PDOX) model has been recently established [85]. Tumor cells are first extracted from patients undergoing resection for colorectal adenocarcinoma and are tagged by luciferase. The collected luciferase-tagged tumor cells are then inoculated with stroma cells to physiologically mimic the tumor microenvironment, followed by being orthotopically injected into the rectum of male NOD/SCID mice. As tumor cells are tagged by luciferase, metastasis can be monitored by an imaging machine. Upon sacrifice, tumors were detected in 96.9% of PDOX mice. Histological examination demonstrated a similarity of architecture between xenografts and the originated tumors in patients, together with the occurrence (53.1%) of metastases in the liver and lung. Importantly, the mortality of PDOX mice was reported to be 0%, indicating that PDOX is a safe and efficient mouse model to recapitulate CRC metastasis.

Currently, the PDOX mouse model has been widely applied in the evaluation of cancer drug efficiency. For example, using PDOX mice, the combination of temozolomide, pazopanib and FOLFOX (oxaliplatin, leucovorin and 5-fluorouracil) was reported to be an effective treatment for CRC through inhibiting lymphangiogenesis [86]. However, the long establishment period of the PDOX model has been an obstacle for its utilization. PDOX mice also lack stromal compounds, including fibroblasts and blood vessels, resulting in the absence of interaction between tumor stroma and the tumor microenvironment [38]. To address these issues, a modified model was recently developed by first co-transplanting patient-derived organoids with endothelial colony-forming cells (ECFC) into NRGA-immunodeficient mice (PDOXwE), which was then sub-cultured in Balb/c mice [87]. PDOXwE mice have fast tumor growth, meanwhile maintaining the primary characteristics of patient-derived organoids. The formation of new blood vessels was also observed in PDOXwE mice. Mechanistically, ECFCs are circulating endothelial progenitor cells contributing to neovascularization in several pathophysiological conditions [88]. These findings thus indicate that the PDOXwE model can recapitulate the tumor microenvironment more accurately than common PDOX mouse models. Additionally, similar sensitivity to chemotherapy drugs between PDOXwE and PDO mice was also observed [87], implying the potential of these mouse models as important tools for the development of individualized therapy. Nevertheless, it is noteworthy to highlight that although genotypic characteristics of the primary tumor in patients are highly preserved in PDOX mice, changes in copy number can still occur after xenotransplantation [38]. Hence, before the popularization of using a PDOX mouse model for drug screening, further investigations are necessary to confirm the consistency between the PDOX result and clinical patient response.

In general, the above models greatly facilitate the investigation of genes that contribute to CRC development and metastasis as well as the selection of most appropriate therapeutic options. However, several limitations still remain to be solved. For instance, the gut microbiota plays a pivotal role in CRC progression, and accumulated evidence has reported the importance of the microbiota in cancer metastasis [89]. Although orthotopic injection of organoids into mice has provided new approach for the study of metastasis, it is still different from the actual metastasis that occurs in human. Extensive studies are required to advance organoid cultures to include more components of the tumor microenvironment, such as immune cells, stroma cells, and intratumoral microbes, thereby establishing organoid models that can more accurately recapitulate the metastatic conditions in human.

### 4.2. Transgenic Mouse Model of Metastasis

In addition to transplantation mouse models, metastasis can also be induced in transgenic mice, although the latency is long with low penetrance. These transgenic mouse models are mainly characterized by hyperplastic lesions and serrated histology of intestinal epithelium [55]. It was reported that heterozygous deletion of *Braf*^LSL–V637E/+^ leads to full progression from serrated hyperplasia, adenoma, and finally to metastatic carcinoma [90]. However, the latency of this transgenic model is long, and the metastasis rate is only 20%. Metastasis was also observed in the *Tp53*^R172H/+^ mouse model, yet again, these transgenic mice exhibit long latency with a low metastasis rate [55,90], whereas another study developed a mouse model with *Kras*^G12V/+^ and *Pten*^fl/fl^ deletion, and these mice have a higher metastasis rate (41%) compared to other transgenic mouse models; however, the latency is still long [91]. Altogether, unless major breakthroughs are achieved, transgenic mouse models remain a minor alternative of xenotransplantation in the study of metastasis.

## 5. Conclusions and Future Perspectives

Mouse models have been used as robust tools in the study of CRC for over 40 years. For example, the AOM-induced CRC mouse model can recapitulate sporadic CRC with many similar features as observed in CRC patients. However, most of the current CRC mouse models can only induce the formation of adenomas but not adenocarcinomas in the distal colon. Given that invasive adenocarcinoma is the main cause of death in CRC patients, it is urgent to develop and popularize mouse models that can develop adenocarcinoma for preclinical studies. Fortunately, advances in Cre-Loxp technology has provided many new opportunities. The CDX2P-Cre-*Apc*^+/LoxP^ mouse model with AOM treatment was recently established, which can develop colon adenocarcinomas with similar histopathological characteristics as in human sporadic CRC. However, a major downside of this mouse model is the deletion of the *Apc* gene in colon epithelial cells; thus, the development of sporadic CRC cannot be achieved. The next goal of researchers should focus on developing mouse models that can recapitulate sporadic human CRC, thereby allowing more accurate evaluation of drug treatment.

*Apc* mutation has been identified in the initial step of colon tumorigenesis, and many CRC mouse models have been established with the *Apc* mutation being the centrality of these models. To date, a variety of transgenic mouse models that target different loci of *Apc* have been developed. However, tumors developed in these models are mostly located in the small intestine instead of the colon. As we all know, tumors in CRC patients are actually located in the colon. and tumorigenesis in human small intestines is rare. Hence, it is important to develop transgenic mice with specific colon tumorigenesis. This would allow for more precise recapitulation of colon tumorigenesis in humans, further facilitating the impact evaluation of drugs and other treatments in CRC patients.

One of the major goals of mouse models is to recapitulate the pathology of human diseases in order to produce accurate models for testing the efficacy and safety of treatments. In the late stage of human CRC, metastasis frequently occurs, leading to high mortality and recurrence. It is therefore urgent to understand the mechanism of metastasis, thereby developing treatments against metastasis for clinical benefits. Currently, the most commonly used mouse model of metastasis is established by transplantation or orthotopic injection of CRC cells or organoids into nude or SCID mice. Although metastases are developed in these models, they cannot accurately represent the real metastasis that occurs in humans due to the lack of multiple components such as immune cells of the tumor microenvironment. Further studies are necessary to establish mouse models that can accurately recapitulate human metastasis, thus facilitating accurate evaluation of drug treatment.

In summary, with these advanced mouse models, improved mechanistic understanding of colon tumorigenesis can be achieved, thereby providing valuable preclinical findings that can be potentially translated into clinical benefits for CRC patients. Nevertheless, the current mouse models still need to be improved to more precisely mimic human CRC.

## Figures and Tables

**Figure 1 biomedicines-10-01710-f001:**
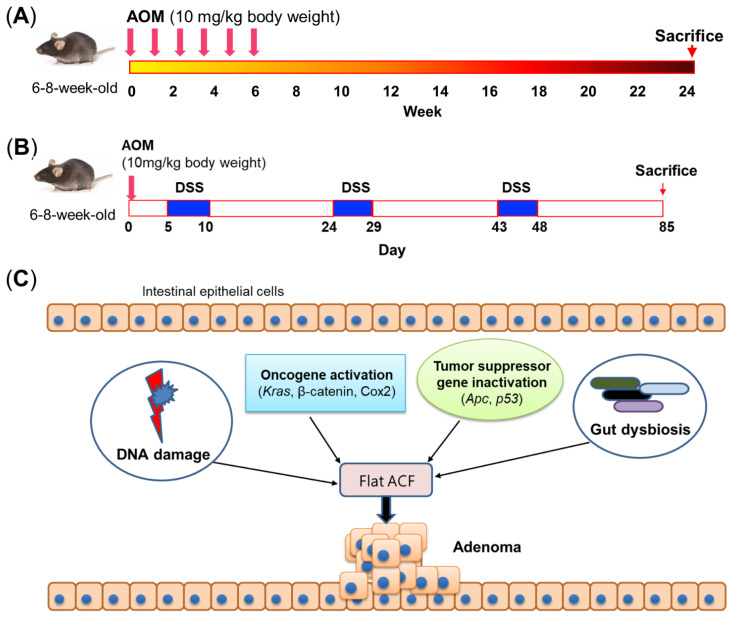
Carcinogen-induced mouse models. (**A**) Schematic of commonly used method of AOM treatment. Mice are provided a total of 6 AOM doses (10 mg/kg intraperitoneal injection per week), and mice can develop colon adenomas about 24 weeks after the last AOM injection. (**B**) Schematic of commonly used method of AOM/DSS treatment. A single intraperitoneal injection of AOM (10 mg/kg body weight) is first provided, followed by three treatment cycles of 1–3% DSS in drinking water for 5 days and then regular water treatment for 14 days. Mice can develop colon tumors about 85 days after AOM injection. (**C**) Mechanism of AOM-induced colon tumorigenesis. AOM can induce DNA damage, activate oncogenes, and inactivate tumor suppressor genes to initiate colon tumorigenesis. In addition, the gut microbiota is also necessary for AOM-induced colon tumorigenesis. ACF, aberrant crypt foci; AOM, azoxymethane; *Apc*, adenomatous polyposis coli; COX2, cyclooxygenase 2; DSS, dextran sulfate sodium; *Kras*, Kirsten rat sarcoma viral oncogene homolog; *p53*, tumor protein *p53*.

**Figure 2 biomedicines-10-01710-f002:**
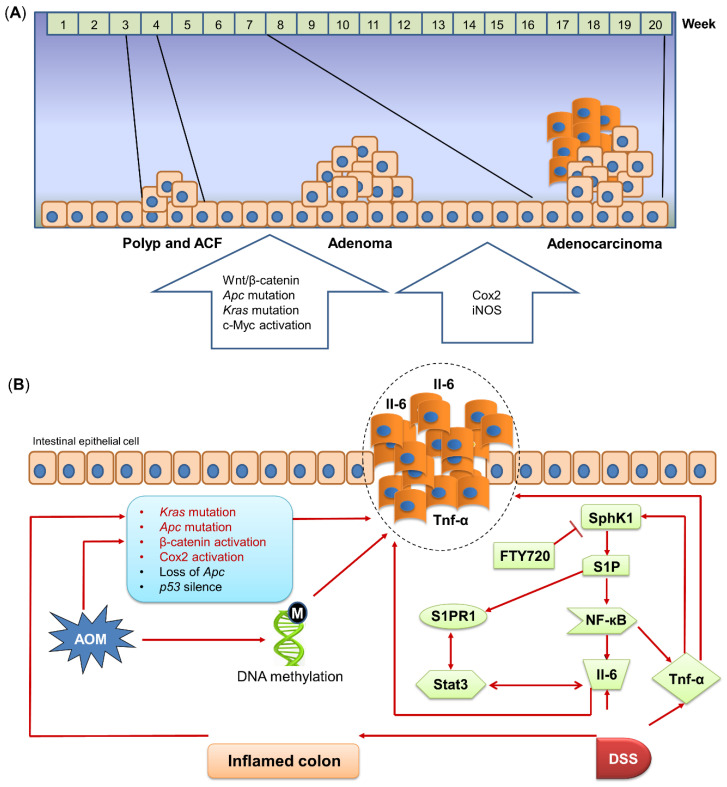
Mechanism of AOM/DSS-induced mouse model. (**A**) Molecular alterations in different stages of AOM/DSS-induced colon tumorigenesis. AOM/DSS treatment induces the progression from polyps/ACFs and adenomas to adenocarcinomas. Different genes are correlated with each stage of AOM/DSS-induced tumorigenesis. (**B**) Mechanism of AOM/DSS-induced colon tumorigenesis. AOM induces DNA methylation and activates or mutates cancer-related pathways to initiate tumorigenesis. DSS treatment upregulates Il-6 and Tnf-α expressions in the colon. SphK1, which is induced by Tnf-α, drives the NF-кB/Stat3 pro-inflammatory pathway to induce inflammation and promote tumorigenesis. AOM, azoxymethane; *Apc*, adenomatous polyposis coli; COX2, cyclooxygenase 2; DSS, dextran sulfate sodium; Il-6, interleukin-6; iNOS, inducible nitric oxide synthase; *Kras*, Kirsten rat sarcoma viral oncogene homolog; NF-кB, nuclear factor kappa-light-chain-enhancer of activated B cells; *p53*, tumor protein *p53*; S1P, sphingosine-1-phosphate; S1PR1, sphingosine 1-phosphate receptor-1; SphK1, sphingosine kinase 1; Stat3, signal transducer and activator of transcription 3; Tnf-α, tumor necrosis factor-alpha; Wnt, wingless-related integration site.

**Figure 3 biomedicines-10-01710-f003:**
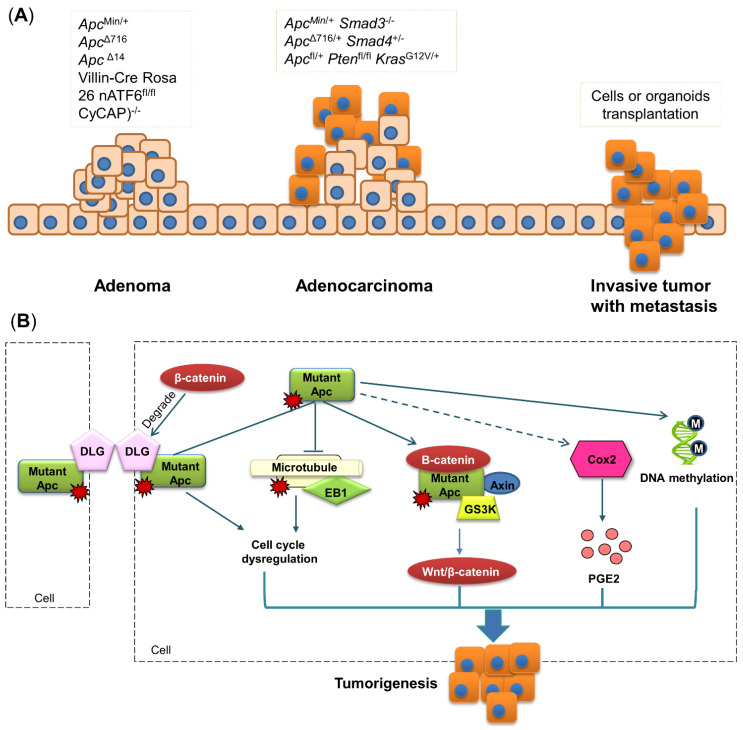
Transgenic mouse models. (**A**) Schematic of gene mutation in each stage of colon tumorigenesis. Single gene mutation mostly induces the development of adenomas, while multiple gene mutations can promote the formation of adenocarcinomas. Metastasis is rarely observed in transgenic mouse models, and its appearance in mice requires transplantation of cancer cells or organoids. (**B**) *Apc* mutation results in the activation of a destruction complex consisting of β-catenin, *Apc*, Axin and GS3K. The EB1-microtubule complex and DLG are destructed by mutant *Apc*, causing cell-cycle dysregulation. *Apc* mutation also promotes DNA methylation and COX2 activation. All these alterations induced by *Apc* mutation can lead to colon tumorigenesis. *Apc*, adeno A recent study matous polyposis coli; COX2, cyclooxygenase 2; CyCAP, cyclophilin C-associated protein; DLG, discs-large; EB1, end-binding protein 1; GSK3, glycogen synthase kinase 3; *Kras*, Kirsten rat sarcoma viral oncogene homolog; nATF6, activating transcription factor 6; PGE2, prostaglandin E2; PTEN, phosphatase and tensin homolog; Wnt, wingless-related integration site.

**Figure 4 biomedicines-10-01710-f004:**
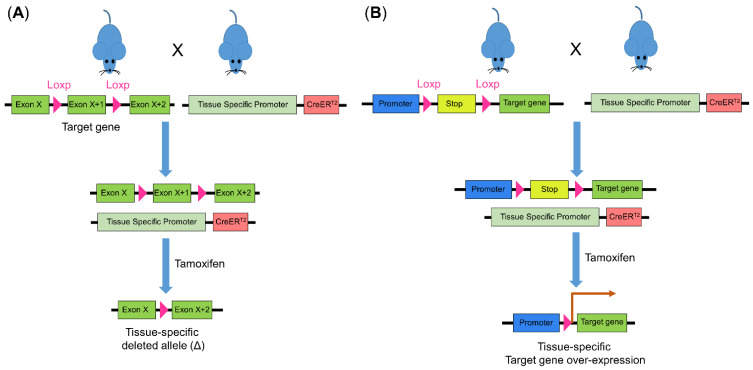
Cre-Loxp recombination system for genetic modulation in mice. The introduction of Loxp to flank the target gene creates a conditional knock-out prerequisite. Mice with the introduced Loxp is inter-crossed with mice carrying Cre whose promoter is tissue specific. Using tamoxifen, Cre is activated and (**A**) leads to target gene deletion or (**B**) deletion of the stop motif and subsequent overexpression of the target gene. Cre, Cre recombinase.

## Data Availability

Data are available within the article.

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
