# Peer review of "Mouse Models for Application in Colorectal Cancer: Understanding the Pathogenesis and Relevance to the Human Condition"

_biomedicines, 2022, doi:10.3390/biomedicines10071710_

Round 1

Reviewer 1 Report

Dear authors,

The new manuscript takes into consideration numerous of my previous remarks, and I appreciate these efforts. Some paragraphs are better constructed and it facilites the understanding. 

However, minors changes are expected: 

-lines 71-72: the stability of AOM is not well referenced. Please, revised it. 

-lines 139-140: the authors refered to the publication 20 to argument the "predominant location of tumors being middle and distal colon" ; I do not have found this information in the paper. Please, correct it. 

-lines 205-206: the authors present KRAS and refered to publication 25; however, this paper deals with WTRas protein that can suppress tumorigenic properties of alternate mutant Ras family members. Please, check this reference. 

-lines 215-261: what is the point of specifying that "codons 33,34,37,41 being G:C to A:T transitions in the tumors of AOM-treated mice" ? 

-lines 263-264: the authors explain that "p53 was also found to be elevated in AOM/DSS-treated mice" with the reference 47. However, this paper deals with feline vaccine. Please, find a better reference to argue. 

-lines 275-277: the authors explain that "since, antibiotics administration could inactivate STAT3 signaling due to the lack of bacteria invasion" and refered to the reference 50. This paper deals with ATF6 implication in CRC development. Please, correct it. 

-lines 373-374: the authors present that "DLG is located at cell-cell contacts where it acts as a scaffold to interact with other proteins". However, the reference 69 deals with the pathway RP-MDM2-p53 that is a critical mediator of CRC following APC loss. Please, correct it. 

-391-393: the reference 75 deals with results obtained in zebrafish. I do not think that this reference is well appropriate. 

-lines 416-417: the authors write that "ATF is a basic leucine zipper transcription factor belo,nging to the CREB and ATF family transcription factyors". I do not think that the reference 70, which deals with ATF6 expression in lesions undergoing pre-cancerous atypical change in non ulcerative colitis (UC) and UC associated with CRC is adapted. Please, correct it with a more adatpted reference. 

-lines 479-480: the authors explain that "about 40% of vascular endothelial cells within the tumor microenvironment are originated from ECFs" with the reference 88. This paper deals with the anti-angiogeneis therapy which targets stem cells. Please, correct it or specifiy. 

Thank in advance to complete your work. 

Author Response

The new manuscript takes into consideration numerous of my previous remarks, and I appreciate these efforts. Some paragraphs are better constructed and it facilites the understanding.

However, minors changes are expected:

1. lines 71-72: the stability of AOM is not well referenced. Please, revised it.

Response: Thanks for the comments. We have now added references of the stability of AOM to Page 2, Line 72.

2. lines 139-140: the authors refered to the publication 20 to argument the "predominant location of tumors being middle and distal colon"; I do not have found this information in the paper. Please, correct it.

Response: We have now replaced the citation on Page 4, Line 139 with the publication 10.

3. lines 205-206: the authors present KRAS and refered to publication 25; however, this paper deals with WT Ras protein that can suppress tumorigenic properties of alternate mutant Ras family members. Please, check this reference.

Response: The citation on Page 7, Line 205 is now replaced with the publication 26.

4. lines 215-261: what is the point of specifying that "codons 33,34,37,41 being G:C to A:T transitions in the tumors of AOM-treated mice"?

Response: β-catenin mutation in mice as well as in humans could occur in different codons with distinct consequences. For specification, we listed the codons that are most frequently mutated in AOM-treated mice.

5. lines 263-264: the authors explain that "p53 was also found to be elevated in AOM/DSS-treated mice" with the reference 47. However, this paper deals with feline vaccine. Please, find a better reference to argue.

Response: We appreciate the suggestion. The citation on Page 8, Line 264 is now replaced with the publication 49 which is a study based on a mouse model of colitis-associated colorectal cancer.

6. lines 275-277: the authors explain that "since, antibiotics administration could inactivate STAT3 signaling due to the lack of bacteria invasion" and refered to the reference 50. This paper deals with ATF6 implication in CRC development. Please, correct it.

Response: The statement on Page 8, Line 276 is now corrected by including ATF6.

7. lines 373-374: the authors present that "DLG is located at cell-cell contacts where it acts as a scaffold to interact with other proteins". However, the reference 69 deals with the pathway RP-MDM2-p53 that is a critical mediator of CRC following APC loss. Please, correct it.

Response: We have now replaced the citation on Page 11, Line 374 with the publication 71.

8. 391-393: the reference 75 deals with results obtained in zebrafish. I do not think that this reference is well appropriate.

Response: The reference 75 and its related statement are now removed from the text.

9. lines 416-417: the authors write that "ATF is a basic leucine zipper transcription factor belo,nging to the CREB and ATF family transcription factyors". I do not think that the reference 70, which deals with ATF6 expression in lesions undergoing pre-cancerous atypical change in non ulcerative colitis (UC) and UC associated with CRC is adapted. Please, correct it with a more adatpted reference.

Response: We have now replaced the citation on Page 12, Line 416 with the publication 79.

10. lines 479-480: the authors explain that "about 40% of vascular endothelial cells within the tumor microenvironment are originated from ECFs" with the reference 88. This paper deals with the anti-angiogeneis therapy which targets stem cells. Please, correct it or specifiy.

Thank in advance to complete your work.

Response: We agree with the reviewer that the reference 88 did not focus on the vascular endothelial cells. To give a more accurate content, the statement on Page 13, Line 477 is now removed.

Reviewer 2 Report

Th authors of the manuscript responded to all raised concerned and made the necessary corrections and editing.

  I do not have any further suggestions.

Author Response

Th authors of the manuscript responded to all raised concerned and made the necessary corrections and editing.

I do not have any further suggestions.

Response: We appreciate the feedbacks of the reviewer.

This manuscript is a resubmission of an earlier submission. The following is a list of the peer review reports and author responses from that submission.

Round 1

Reviewer 1 Report

The review by Li C and co-authors provides the analysis of the available approaches for generation of the mouse models for studying the mechanisms of colon carcinogenesis . The manuscript is thoroughly written and discusses the important aspects of animal models, including  the pros and cons of the different models and mechanisms of colon carcinogenesis tested using animal models. The review provides a concise and timely analysis of the CRC animal models and will be a useful addition to a rich field of colorectal cancer research.

There are some minor points, which I recommend to address:

1-Line 78, references 13, 14. Authors site their own research publications from 2019 and 2020 about the induction of the colon carcinogenesis using carcinogen AOM, but the publications, in which the protocol was originally reported have to be cited instead.

2-line 169: remove “Whereas “

Line 173- format citation 30, 31.

Line 177- replace the word Mechanically with Mechanistically

Line 184- remove “Whereas”

Line 208- describe abbreviation CAG

Line 334 -replace wheres relatively fewer with “with a few”

Line 335.  Please correct the sentence. 

Although the Apc Min mice share genetic similarities, phenotypically they are not exactly recapitulated the FAP disease. They mostly develop the intestinal adenomas in contrast to the FAP patients, which develop predominantly colon lesions.

Line 368-369. Please correct the sentence to … induce mostly formation of adenomas, but not invasive tumors,

Line 385. The reference 53, cited here, is incorrect.  Please use the correct reference.  

Author Response

The review by Li C and co-authors provides the analysis of the available approaches for generation of the mouse models for studying the mechanisms of colon carcinogenesis. The manuscript is thoroughly written and discusses the important aspects of animal models, including the pros and cons of the different models and mechanisms of colon carcinogenesis tested using animal models. The review provides a concise and timely analysis of the CRC animal models and will be a useful addition to a rich field of colorectal cancer research.

There are some minor points, which I recommend to address:

1. Line 78, references 13, 14. Authors site their own research publications from 2019 and 2020 about the induction of the colon carcinogenesis using carcinogen AOM, but the publications, in which the protocol was originally reported have to be cited instead.

Response: Thanks for the comments. We have now changed the citations for the original protocol (Page 5, Line 2).

2. Line 169: remove “Whereas “

Response: We have now removed “Whereas” as suggested (Page 8, Line 7).

3.Line 173- format citation 30, 31.

Response: Citation 30, 31 are now reformatted (Page 8, Line 12).

4. Line 177- replace the word Mechanically with Mechanistically

Response: We have now replaced the word with “Mechanistically” (Page 8, Line 16).

5.Line 184- remove “Whereas”

Response: We have now removed the word as suggested (Page 8, Line 22).

6. Line 208- describe abbreviation CAG

Response: We have now included the full name of CAC (colitis-associated CRC) on Page 9, Line 23.

7. Line 334 -replace wheres relatively fewer with “with a few”

Response: We have now changed the term to “with a few” on Page 14, Line 3.

8. Line 335. Please correct the sentence.

Response: The sentence on Page 14, Line 4 is now changed to “ApcMin/+ mice have many genetic and phenotypic similarities to patients with FAP with 90% identical in orthology.”.

9. Although the Apc Min mice share genetic similarities, phenotypically they are not exactly recapitulated the FAP disease. They mostly develop the intestinal adenomas in contrast to the FAP patients, which develop predominantly colon lesions.

Response: We have now added a statement to highlight that ApcMin/+ mice mostly develop intestinal adenomas in contrast to FPA patients which predominantly colon lesions (Page 14, Line 5).

10. Line 368-369. Please correct the sentence to … induce mostly formation of adenomas, but not invasive tumors.

Response: The sentence on Page 14, Line 24 is now corrected as suggested.

11. Line 385. The reference 53, cited here, is incorrect. Please use the correct reference.

Response: We have now corrected the reference on Page 15, Line 16.

Reviewer 2 Report

Dear authors,

This work deals with interesting research area in the field of colorectal cancer models. However, this topic has already been described numerous times.

It is disappointing that authors do not refered to the most recent articles (as Bürtin et al, 2020 "Mouse models of CRC: past, present and future perspectives" or Stastna et al 2019 "Human CRC from the perspectives of mouse moldels"). 

A minor comment concernes some references that do not share the adapted style of the journal throughout the text (26, 60, 80, 100 and 104). 

Some paragraphs have to be improved. Instead of making a catalog of  bibliographical data, authors should cited less references but should used the main references with more accuracy and more than just one time.  

My major comment concern the data reported in this work: some references are not correctly used and do not refered to their main results. In my opinion, a review has to resume and simplify numerous works and this challenge is not reached in this manuscript. 

Finally, authors should have detailed the most recent models as PDX and/or organoids that are only explained by 6 references (104 to 110).  

For all these reasons, this manuscript is not ready to be accepted with this actual form. 

Author Response

This work deals with interesting research area in the field of colorectal cancer models. However, this topic has already been described numerous times.

1. It is disappointing that authors do not refered to the most recent articles (as Bürtin et al, 2020 "Mouse models of CRC: past, present and future perspectives" or Stastna et al 2019 "Human CRC from the perspectives of mouse moldels").

Response: We appreciate the comments. The suggested articles are now cited in Introduction (Page 3, Line 21).

2. A minor comment concerns some references that do not share the adapted style of the journal throughout the text (26, 60, 80, 100 and 104).

Response: We have now revised all references throughout the manuscript to ensure that they are correctly cited with the adapted style of the journal.

3. Some paragraphs have to be improved. Instead of making a catalog of bibliographical data, authors should cite less references but should use the main references with more accuracy and more than just one time.

Response: The whole manuscript is now revised to improve the contents. We have also reorganized our citations to remove some references and use the main reference more accurately.

4. My major comment concern the data reported in this work: some references are not correctly used and do not refered to their main results. In my opinion, a review has to resume and simplify numerous works and this challenge is not reached in this manuscript.

Response: All references are now revised to ensure that they are correctly used and refer to their main results. We have also removed some references to simplify citations in the manuscript.

5. Finally, authors should have detailed the most recent models as PDX and/or organoids that are only explained by 6 references (104 to 110).

Response: We have now added more discussion on the most recent mouse models with xenotransplantation as following:

(Page 20, Line 9)

Currently, PDOX mouse model has been widely applied in the evaluation of cancer drug efficiency. For example, using PDOX mice, the combination of temozolomide, pazopanib and FOLFOX (oxaliplatin, leucovorin and 5-fluorouracil) was reported to be an effective treatment for CRC through inhibiting lymphangiogenesis. However, the long establishment period of PDOX model has been an obstacle for its utilization. PDOX mice are also lack of stromal compounds including fibroblasts and blood vessels, resulting in the absence of interaction between tumor stroma and tumor microenvironment. To address these issues, a modified model was recently developed by first co-transplanting patient-derived organoids with endothelial colony-forming cells (ECFC) into NRGA immunodeficient mice (PDOXwE), which was then sub-cultured in Balb/c mice. PDOXwE mice have fast tumor growth, meanwhile maintaining the primary characteristics of patient-derived organoids. The formation of new blood vessels was also observed in PDOXwE mice. Mechanistically, ECFCs are circulating endothelial progenitor cells contributing to neovascularization in several pathophysiological conditions, of which about 40% of vascular endothelial cells within the tumor microenvironment are originated from ECFCs. These findings thus indicated that PDOXwE model could recapitulate the tumor microenvironment more accurately than common PDOX mouse models. Additionally, similar sensitivity to chemotherapy drug between PDOXwE and PDO mice was also observed, implying the potential of these mouse models as important tools for the development of individualized therapy. Nevertheless, it is noteworthy to highlight that although genotypic characteristics of the primary tumor in patients are highly preserved in PDOX mice, changes in copy number could still occur after xenotransplantation. Hence, before the popularization of using PDOX mouse model for drug screening, further investigations are necessary to confirm the consistency between PDOX result and clinical patient response.

6. For all these reasons, this manuscript is not ready to be accepted with this actual form.

Response: We have now improved the manuscript by adding more contents and reorganizing the citations based on the reviewer suggestions.

Round 2

Reviewer 2 Report

Dear authors,

Even if you made some changes in the new manuscript, it still contains mistakes that prevent my approval for accepting it.

For example, you prefer to add new references instead of selectioning the more appropriate and to re-write paragraphs. 

Numerous misinterpretation are present in the manuscript and as I mentioned previously, it is not acceptable. One example is for the reference 107 which deals with bioluminescence imaging for tumor growth and metastasis monitoring; in which "high distant organ metastasis rates were 33.3% detected liver or lung metastasis for UCC (urothelial cell-carcinoma) and 53.1% for CRC (colorectal cancer)". Unfortunately, you report that for colorectal cancer "Upon sacrifice, tumors in PDOX mice were found to be capable of resembling the characteristics of originated tumors in patients, together with the occurrence of metastases in the liver (33.3%) and lung (53.1%)." This example demonstrates that you misinterprete data from literature, and this type of errors is frequent along the text. 

This manuscript is not yet suitable to be accepted.